# Ice core site considerations from modeling $CO_2$ and $O_2/N_2$ ratio diffusion in interior East Antarctica

Marc J. Sailer<sup>1</sup>, Tyler J. Fudge<sup>1</sup>, John D. Patterson<sup>2</sup>, Shuai Yan<sup>1</sup>, Duncan A. Young<sup>3</sup>, Shivangini Singh<sup>3</sup>, Don Blankenship<sup>3</sup>, Megan Kerr<sup>3</sup>

Correspondence to: Marc J. Sailer (marc.sailer@colorado.edu) or Tyler J. Fudge (tjfudge@uw.edu)

Abstract. Obtaining a continuous ice core record to 1.5 million years (Ma), which spans the Mid-Pleistocene Transition (MPT, 1.2 to 0.7 Ma) is a goal of multiple international efforts in Antarctica. Ice of such age is likely to be highly thinned and located in warm ice near the bed, conditions which promote diffusion of the stored atmospheric gases. Here, we assess the preservation of CO<sub>2</sub> and the O<sub>2</sub>/N<sub>2</sub> ratio in the ice sheet between South Pole and Dome A where the NSF Center for Oldest Ice Exploration has surveyed with airborne radar. We employ two models: 1) a 1D, steady state ice and heat flow model to calculate the temperature and age of ice with respect to depth, and 2) a vertical gas diffusion model for clathrate ice. We analyze the preservation of CO<sub>2</sub> signals with a period of 40 kyr to match pre-MPT glacial cycles and the preservation of O<sub>2</sub>/N<sub>2</sub> signals with a period of 20 kyr to match precession cycles. 1.5 Ma ice is lost to basal melt in much of the study area where ice thickness exceeds 3000 m. In locations that preserve 1.5 Ma ice, vertical gas diffusion is most sensitive to accumulation rate; high accumulation rate sites have more highly thinned old ice, and the steeper gas concentration gradients enhance diffusion. The most promising region for recovering 1.5 Ma ice is approximately 400 km from both South Pole and Dome A, a region we call the "Foothills," due to low accumulation rates and moderate ice thickness. CO<sub>2</sub> signals lose on average 14 % of their amplitude, while O<sub>2</sub>/N<sub>2</sub> signals lose on average 95 % for 1.5 Ma ice, suggesting precession cycles may not be identifiable. Unknown geothermal heat flow is a large uncertainty for both ice loss from basal melt and gas signal preservation.

#### 1 Introduction

15

Extending the continuous ice core record through the Mid-Pleistocene Transition (MPT) to ~1.5 million years (Ma) is a goal of the International Partnership in Ice Core Sciences (Fischer et al., 2013). The MPT is a climatic transition that occurred between 1.2 and 0.7 Ma during which Earth's glacial cycles shifted from high frequency (~40 thousand years, kyr), low amplitude events to low frequency (~100 kyr), high amplitude events (Head et al., 2008). The MPT has been observed in the benthic  $\delta^{18}$ O record (Figure 1, top), a climate proxy used to estimate global ice volume (Lisiecki and Raymo, 2005); the cycle transition was first noted by Shackleton and Opdyke (1976). The mechanisms underlying the MPT are not fully understood (Clark et al., 2006; Tziperman and Gildor, 2003; Rial, 2004). The role of a proposed gradual decline in atmospheric CO<sub>2</sub> remains an open question (Hönisch et al., 2009; Martin et al., 2024). Atmospheric CO<sub>2</sub> concentrations prior to the ice core

<sup>&</sup>lt;sup>1</sup>Department of Earth and Space Sciences, University of Washington, Seattle, 98195, United States

<sup>&</sup>lt;sup>2</sup>Department of Earth System Sciences, University of California Irvine, Irvine, 92697, United States

<sup>&</sup>lt;sup>3</sup>Institute for Geophysics, University of Texas, Austin, 78758, United States

record limit (0.8 Ma; EPICA, 2004) have been reconstructed using  $\delta^{11}B$  derived from planktic foraminifera (Bartoli et al., 2011; Chalk et al., 2017; de la Vega et al., 2020; Dyez et al., 2018; Guillermic et al., 2022; Henehan et al., 2013; Hönisch et al., 2009; Martínez-Botí et al., 2015; Seki et al., 2010). However, this method is limited, providing different results based on the species and location and requiring several assumptions. Köhler (2023) suggests that some of these assumptions, such as equilibrium between atmospheric and equatorial sea surface pCO<sub>2</sub>, lower estimated surface ocean pH in the Pacific than the Atlantic, and assumptions on total alkalinity and dissolved inorganic carbon, may be incorrect by comparing reconstructions to a carbon cycle model. Therefore, answering questions regarding atmospheric CO<sub>2</sub> concentration's role in the MPT may require direct measurements from an ice core older than 0.8 Ma.

Figure 1: (Top) Record of benthic  $\delta^{18}O$ , a proxy for global ice volume, from the LR04 Stack (Lisieki and Raymo, 2005). The MPT (top magenta) is the transition from low amplitude, high frequency glacial cycles to high amplitude, low frequency cycles. The ice core record of  $CO_2$  (middle: Ahn et al., 2012; Bereiter et al., 2015; Lüthi et al., 2008; Indermühle et al., 2000; Siegenthaler et al., 2005; see National Centers for Environmental Information: Antarctic Ice Cores Revised 800 KYr CO2 Data website for data) does not cover the full MPT. An extended ice core record of  $CO_2$  back to 1.5 Ma may provide insights about the MPT. The  $O_2/N_2$  ratio in the ice core record (bottom red; Bouchet et al., 2023; see Bouchet et al., 2023 for data), which has been observed to follow maximum summer insolation at 75° S (bottom grey). This can be used for dating an extended ice core by matching precession cycles in the  $O_2/N_2$  ratio. Time axis is inverted for all plots.

50

The National Science Foundation Center for Oldest Ice Exploration (NSF-COLDEX) is investigating the area between the South Pole and Dome A to identify a drill site for a continuous deep ice core reaching 1.5 million years (Young et al., 2025). Ice of this age will have thinned significantly, steepening gas concentration gradients. Furthermore, it will have experienced a long duration of warm temperatures near the bed, increasing the diffusivity and solubility of the gases. Thus, both steeper

gradients and warmer temperature promote diffusive smoothing in trapped gases that may inhibit recovery of atmospheric gas records, where 1.5 Ma ice is recovered.

In this study, we assess the preservation of atmospheric gas signals in 1.5-million-year-old ice. We first model the depth-age and temperature profiles in the COLDEX study area and then model the diffusion of CO<sub>2</sub> and the O<sub>2</sub>/N<sub>2</sub> ratio. We analyze the effects of accumulation rate, ice thickness, surface temperature, and geothermal heat flow (GHF) on gas signal preservation.

Accumulation rate has opposing effects on gas signal preservation. High accumulation rates advect cold surface temperatures deeper into the ice column, leading to a cooler temperature profile. On the other hand, high accumulation rates lead to increased thinning in near-basal ice layers, where old ice is present, and thus increases gas concentration gradients. Ice thickness also has opposing effects on gas signal preservation. Thicker ice columns cause less thinning, allowing for thicker layers and less steep gradients in near-basal ice. However, a thick ice column also increases the basal temperature and leads to more basal melt, potentially removing the oldest ice. Surface temperature and GHF are boundary conditions for the temperature profile, so lower values will result in a cooler temperature profile. Moreover, minimizing near-basal temperatures will mitigate the basal melt rate, increasing the likelihood that old ice is present.

We focus on CO<sub>2</sub> and the O<sub>2</sub>/N<sub>2</sub> ratio in this study. CO<sub>2</sub> is an important paleoclimate parameter (Ahn et al., 2012; Bereiter et al., 2015; Lüthi et al., 2008; Indermühle et al., 2000; Siegenthaler et al., 2005). Long-term cooling, often attributed to an assumed secular decrease in atmospheric pCO<sub>2</sub>, is at the heart of many possible explanations for the MPT (Raymo, 1997; Mudelsee and Schulz, 1997; Berger et al., 1999). The O<sub>2</sub>/N<sub>2</sub> ratio is an important dating tool (Kawamura et al., 2007; Bouchet et al., 2023) because it records precession cycles by tracking maximum summer insolation, due to influences on snow metamorphism and grain properties in shallow firn (Martinerie et al., 1992; Bender et al., 1994; Bender et al., 1995; Bender 2002; Kawamura et al., 2007; Fujita et al., 2009; Lipenkov et al., 2011). This is a key method in dating gases in the oldest ice cores (Oyabu et al., 2022). Histories of these gases from the current ice core record can be found in Fig. 1.

We build upon the work of Bereiter et al. (2014) where the preservation of CO<sub>2</sub> and the O<sub>2</sub>/N<sub>2</sub> ratio is evaluated for an idealized ice core location. Here, we similarly evaluate signal preservation using the new information provided by the NSF-COLDEX airborne survey (Young et al., 2025). We limit our discussion of CO<sub>2</sub> and the O<sub>2</sub>/N<sub>2</sub> ratio to the periods of 40 and 20 kyr, respectively, to match pre-MPT glacial cycles for CO<sub>2</sub> and precession cycles for the O<sub>2</sub>/N<sub>2</sub> ratio. Bereiter et al. (2014) modeled O<sub>2</sub>/N<sub>2</sub> diffusion under two sets of gas parameter values, a "fast set" after Ikeda-Fukazawa et al. (2005) and a "slow set" after Salamatin et al. (2001). Here, we consider only the "slow set" due to the unrealistic modeling results obtained by Oyabu et al. (2021) when using the fast set parameters (Sect. 2.3).

#### 2 Methods

The diffusive smoothing that atmospheric signals stored in the ice sheet have experienced depends on both the temperature history of the ice and the gradient in gas concentrations. These parameters are determined by the time-evolving depth-age

relationship. In this section, we first describe the study area and data sources for model forcings. We then describe a one-dimensional, steady-state model of ice and heat flow (Fudge et al., 2019; Firestone et al., 1990; Lliboutry, 1979). This model is used to provide the physical parameters for a gas diffusion model (Bereiter et al., 2014).

## 90 **2.1 Study area**

100

105

110

The COLDEX study area is divided into three regions based loosely on the ice thickness and its variability (Figure 2):

- Foothills upstream-most portion of the survey towards Dome A with variable ice thickness that typically ranges from ~2500 to ~3000 m.
- Deep basin the middle of the survey area where ice thickness typically exceeds ~3000 m.
- Saddle region the region upstream from South Pole where ice thickness becomes more moderate, ~2000 m to ~3000 m, and the ice flow diverges.

The accumulation rates across the study area are inferred from an englacial layer dated to 4.7 ka and given in ice equivalent (https://doi.org/10.15784/601994). Because we use a steady-state model, we scale the inferred accumulation rate for the past 4.7 ka to the long-term (past four glacial cycles, 450 ka) average at Dome C using the AICC2012 chronology (Bazin et al., 2013; Veres et al., 2013). The average accumulation rate for the past 4.7 ka is 2.9 cm yr<sup>-1</sup> and for the past 450 ka is 2.1 cm yr<sup>-1</sup>. This gives a factor of 0.72. Figure 8a shows the accumulation rate across the study area. Surface temperatures, which also approximate a glacial-cycle average, are interpolated across the region based on the easting from the South Pole, with the warmest temperatures, -54 °C, at the South Pole, cooling linearly to the grid-east to the Dome A foothills, reaching -60 °C at the end of the study area (Figure 8c). We chose -54 °C at the South pole from a rough time weighted average temperature between glacial and interglacial periods (Buizert et al., 2021; Kahle et al., 2021) and -60 °C at the foothills assuming a combination of lapse rate from the 800 m elevation gain from the South Pole and warmer northward temperatures. GHF is poorly known in the study region because there are no direct measurements. Remotely sensed estimates (Stål et al., 2020; Shen et al., 2020; An et al., 2015; Fox-Maule et al., 2005; Martos et al., 2017; Shapiro and Ritzwoller, 2004; Purucker, 2013; Hazzard and Richards, 2024) vary widely (e.g. van Liefferinge et al., 2018). Therefore, we assume a uniform GHF across the study area and consider a range of values to assess how uncertainty in GHF values may affect near-basal ice age, melt rate, and signal diffusion.

Figure 2: (a) COLDEX study area and sub-regions used for this study, defined loosely on ice thickness (see Young et al., 2025, and Bedmap3 for data). White lines indicate flight paths. Grey contour lines represent 100 m elevation change (see REMA v2 for data). (b) Map of East Antarctica. The red rectangle represents the study area. Contour lines represent 500 m elevation change.

## 2.2 Temperature and age-depth model

120

125

We use a one-dimensional, steady state ice and heat flow model to calculate the temperature and age of ice with respect to depth. The age model is identical to that in Bereiter et al. (2014). The temperature model differs in that we utilize a temperature dependent thermal conductivity and specific heat capacity, rather than an ice column average. If the basal temperature exceeds the pressure melting point, the melt rate is calculated. We choose a steady-state model because the conditions near the base of the ice sheet, where the vast majority of diffusion occurs, are relatively insensitive to the transient surface climate forcing. In a set of ice flow model runs, glacial-interglacial changes in surface temperature of 8 °C results in a difference in basal temperature of <0.5 °C (Buizert et al., 2021). This model utilizes input parameters of accumulation rate, ice thickness, surface temperature, GHF, and a parameter for the vertical deformation profile (Lliboutry, 1979). Figure 3a, b, c, and d show an example output of this model using forcings of 2 cm yr<sup>-1</sup> accumulation rate, 3000 m ice thickness, -60 °C surface temperature, 50 mW m<sup>-2</sup> GHF, and a vertical deformation parameter of p = 4.

The steady heat equation, following Firestone et al. (1990), is solved to obtain the vertical temperature profile:

$$\frac{\partial^2 T}{\partial z^2} - \frac{w(z)}{\kappa} \frac{\partial T}{\partial z} = 0, \tag{1}$$

where T is temperature, z is height above the bed. The thermal diffusivity is given by  $\kappa = \frac{\kappa}{\rho c}$ , where  $\rho$  is the constant firm column density based on South Pole values (Lilien et al., 2018), c is the temperature-dependent specific heat capacity given by Cuffey and Paterson (2010; eq 9.1), and k is the thermal conductivity exponentially fit to the Waite et al. (2006) data as

described in the supplementary information of Buizert et al. (2021) (Specific values can be found in the setup\_thermal\_herc.m file in the GitHub repository). w is the vertical velocity as calculated by Parrenin et al. (2007b):

$$w(z) = -(A - m)\psi(z) - m, \tag{2}$$

$$\psi(z) = 1 - \frac{p+2}{p+1} \left( 1 - \frac{z}{H} \right) + \frac{1}{p+1} \left( 1 - \frac{z}{H} \right)^{p+2},\tag{3}$$

where A is accumulation rate, H is ice thickness, and p is a tuning parameter for vertical deformation (Lliboutry, 1979). The deformation parameter is assumed to be p=4 in all scenarios, unless otherwise stated. We include the bedrock in the thermal calculation and set the GHF at 7 km below the ice surface. The melt rate, m, is first set at 0 for calculating the temperature profile. Equation (1) is solved similarly to Firestone et al. (1990) with an integration factor:

$$F(z) = exp\left[-\int_{0}^{z} \frac{w(\zeta)}{\kappa} d\zeta\right]. \tag{4}$$

This gives the following solution:

$$T(z) = T_S - \frac{Q}{K} \int_H^z \frac{1}{F(\eta)} d\eta. \tag{5}$$

- where  $T_S$  is the surface temperature and Q is GHF. The melt rate is calculated from the excess geothermal heat that the ice cannot conduct away from the bed. This is calculated iteratively because the melt rate affects the vertical velocity (equation 2) and thus the englacial temperature profile. The GHF is equal to the heat conduction through the basal ice and the latent heat lost in melting, which we assume is lost as the water flows away from the bed. The example temperature profile in Fig. 3a shows how temperature increases roughly linearly with depth approaching the bed as a result of this solution.
- 150 The vertical age profile is found by solving Equations (2) and (3) with the inferred melt rate and integrating the inverse of the negative vertical velocity:

$$age(z) = \int \frac{-1}{w(z')} dz'. \tag{6}$$

The example age-depth profile in Fig. 3b shows how the age of ice evolves with depth up to 2 Ma, after which older ice has been melted by the warm temperatures shown in Fig. 3a.

#### 155 **2.3** Gas diffusion model


The gas diffusion model uses the temperature- and age-depth profiles described above to calculate the rate of diffusion through time (Ikeda-Fukazawa et al., 2005; Bereiter et al. 2014). Vertical gas diffusion is modeled as follows:

$$\frac{\partial C_{n}}{\partial t} = \frac{\partial}{\partial z} \left( D_{n} \frac{\partial C_{n}}{\partial z} \right), \tag{7}$$

where  $C_n$  is the concentration in the ice of n (CO<sub>2</sub> or O<sub>2</sub>) and  $D_n$  is the diffusion coefficient in ice. Gas exchange with surrounding ice is calculated according to Ikeda-Fukazawa et al. (2005) and Bereiter et al. (2009). The model does not include a portion for diffusion in bubbly ice and instead treats the whole ice column as clathrate ice for simplicity. While diffusion

does occur within bubbly ice, the time gases spend in the bubble phase is relatively short compared to the timescales of interest here (e.g. ice 1000 m below the surface at EDC is ~65 ka, roughly the extent of bubbly ice; Bazin et al., 2013; Veres et al., 2013). We only simulate the diffusion of  $O_2$  because  $N_2$  has a permeation coefficient roughly one order of magnitude lower than that of  $O_2$  (Salamatin et al., 2001). Modeling  $O_2$  diffusion is sufficient for modeling the damping of the  $O_2/N_2$  ratio. Including  $N_2$  diffusion would reduce the smoothing of  $\delta O_2/N_2$  from  $O_2$  diffusion, provided that the gas concentration gradient of  $N_2$  has the same sign as  $O_2$ . This condition is likely met as millennial-scale variable of  $\delta O_2/N_2$  and total air content in ice is thought to be driven by similar bubble-closure processes. This is supported by empirical data showing  $\delta O_2/N_2$  and total air content covariance (Fujita et al., 2009; Lipenkov et al., 2011). Thus, this study provides an upper limit on  $\delta O_2/N_2$  smoothing in ice cores. Note that if  $O_2$  and  $N_2$  gas concentration gradients were of opposite signs, the effects of their diffusion would be additive and thus enhance smoothing.

The concentration of dissolved molecules of gas in pure clathrate ice,  $C_n$ , is calculated as follows:

$$C_n = Z_n \cdot P_n^d \cdot S_n, \tag{8}$$

where  $Z_n$  is the molar fraction of gas n in air enclosed in the clathrate,  $P_n^d$  is the clathrate dissociation pressure of gas n, and  $S_n$  is the solubility of gas n. The clathrate dissociation pressure is species and temperature dependent, and has been experimentally determined (O<sub>2</sub>: Kuhs et al., 2000; CO<sub>2</sub>: Miller, 1961).

$$log_{10}P_n^d = a_n - \frac{b_n}{T},\tag{9}$$

where T is the temperature in Kelvin.







After Bereiter et al. (2014), we use solubility and diffusivity parametrizations from Ahn et al. (2008) and Salamatin et al. (2001) for CO<sub>2</sub> and O<sub>2</sub>, respectively. Diffusive smoothing is dependent on the product of diffusivity and solubility: permeability. Permeability values in clathrate ice are difficult to measure in a lab setting due to their extremely small values, so they are instead indirectly estimated based on a model which simulates air fractionation and diffusive smoothing in an ice core. The coefficients for CO<sub>2</sub> have experimental support at 250 K and are further extrapolated (Ahn et al., 2008). This is the only reliable estimate available for CO<sub>2</sub> permeability, based on an analysis of refrozen melt water in the Siple Dome ice core; however, these estimates have not been independently experimentally verified. Bereiter et al. (2014) modeled O<sub>2</sub>/N<sub>2</sub> diffusion under two sets of gas parameters, a "fast set" and "slow set." The "slow set" parameters are derived from empirical fits to ice core data (Salamatin et al., 2001). The "fast set" parameters are based purely on molecular dynamics simulations of gas diffusion in ice (Ikeda-Fukuzawa et al., 2004). The values for the "fast set" and the "slow set," as well as for CO<sub>2</sub> permeability, can be found in Figure S1 in the Supplement. Oyabu et al. (2021) demonstrates that the "fast set" parameters yield unrealistically high rates of smoothing compared to high resolution O<sub>2</sub>/N<sub>2</sub> measurements from the Dome Fuji core (DF1). Thus, we utilize the "slow set" in this study. For all gases, the permeation coefficients increase as temperature increases, with O<sub>2</sub> increasing more rapidly than CO<sub>2</sub>, but their true sensitivity to temperature is not well known.

Within the gas diffusion model, we calculate the diffusivity as follows:

$$D_n(T) = D_n^0 \exp\left(-\frac{Q_n}{R \cdot T}\right),\tag{10}$$

where  $Q_n$  is the activation energy for the diffusion coefficient of molecules of gas n in ice  $D_n(T)$ , R is the ideal gas constant, T is the temperature, and  $D_n^0$  is a specific constant for each gas (Bereiter et al., 2009; Ikeda-Fukazawa et al., 2005). For CO<sub>2</sub>, we calculate solubility as follows:

$$S_{CO_2}(T) = S_{CO_2}^0 \exp\left(-\frac{E_{CO_2}}{R \cdot T}\right),\tag{11}$$

where  $E_{CO_2}$  is the activation energy for the temperature dependent solubility  $S_{CO_2}(T)$  and  $S_{CO_2}^0$  is a specific constant (Bereiter et al., 2009). To calculate solubility for  $O_2$ , we first calculate the permeability:


$$P_{O_2}(T) = \left[ P_{O_2}^0 exp\left( -\frac{Q_P}{R \cdot T} \right) \right] / P_{diss}^{220K}, \tag{12}$$

where  $Q_P$  is the activation energy for permeation,  $P_{O_2}^0$  is a specific constant (Salamatin et al., 2001), and  $P_{diss}^{220K}$  is the dissociation pressure of  $O_2$  evaluated at 220 K, as in Equation 9. Solubility is then solved as the quotient of the permeability and diffusivity.

The ice column is partitioned into discrete intervals of 50 kyr of ice age. An ice particle being advected down the ice column spends 50 kyr in each interval. An average annual layer thickness and average temperature are calculated for each interval. Gas concentrations are initialized, and the model simulates diffusion based on the properties of the first 50 kyr interval. Next, gas concentrations are saved, and the model physical parameters are updated using the average layer thickness and temperature from the next interval. Then, the simulation continues for the next 50 kyr interval. The process iterates until the last interval is reached (e.g. Figure 3e and f).

We use a 40 kyr period for  $CO_2$  to match the predicted frequency of glacial cycles prior to the MPT and a 20 kyr period for the  $O_2/N_2$  ratio to match precession cycles. To express the amount of signal amplitude lost to diffusion, we define the signal damping ratio (SDR) as:

$$SDR(age) = 1 - \frac{X(age)}{X(0)},\tag{13}$$

where *X(age)* is the gas signal amplitude at a given age and *X(0)* is the initial gas signal amplitude. We use an initial amplitude of 50 ppm for CO<sub>2</sub> and 5 ‰ for the O<sub>2</sub>/N<sub>2</sub> ratio. The SDR gives the fraction of gas signal lost at a target age, with higher values indicating more diffused gas signals. The SDR for 1.5 Ma will be of most interest in this study as it fully covers the MPT and some time prior to it. Figure 3e and f shows how CO<sub>2</sub> and O<sub>2</sub>/N<sub>2</sub> ratio concentrations evolve through time given the corresponding temperature- and age- depth profiles. CO<sub>2</sub> signal amplitudes do not change significantly in this example (Figure 3e). O<sub>2</sub>/N<sub>2</sub> ratio signal amplitudes dampen in old ice, particular in ice older than 0.9 Ma (Figure 3f).

Figure 3: Example output from models using forcings of 2 cm yr<sup>-1</sup> accumulation rate, 3000 m ice thickness, 50 mW m<sup>-2</sup> GHF, and -60 °C surface temperature. (a) Temperature and (b) age profile calculated with 1D, steady state model. (c) Age-temperature relationship. (d) Age-layer thickness relationship. (e) CO<sub>2</sub> and (f) O<sub>2</sub>/N<sub>2</sub> ratio amplitude evolution through time. SDR represents the fraction of gas signal lost. In this example, CO<sub>2</sub> loses very little signal amplitude, only 3 % as indicated by the SDR. The O<sub>2</sub>/N<sub>2</sub> ratio loses much more signal, 80 % by 1.5 Ma, but only begins losing its amplitude around 0.9 Ma.

#### 3 Results



We conduct three sets of simulations with the gas diffusion model. First, we force the model with physical parameters from EPICA Dome C and Dome Fuji to assess whether the existing ice core record can be used to predict the expected diffusion in older records. Second, we assess the sensitivity of each gas's SDR to the four model input parameters to gain an intuition of which forcings are most important. Third, we run the model over the NSF-COLDEX study area to understand where old ice may reside in the region and where each gas will be best preserved.

#### 3.1 Diffusion in existing ice core records

To determine the extent to which ice core gas concentration measurements can be used to estimate the expected diffusion of atmospheric gases in 1.5 Ma ice, we conduct four model runs. In the first run, we simulate the conditions for EPICA Dome C (EDC) and use parameter values from Bereiter et al. (2014) ("EDC" in Table 1). In the second run, we simulate the conditions for Dome Fuji with parameters values tuned to fit the age (Uemura et al., 2018) and borehole temperature data (Buizert et al., 2021) ("DF" in Table 1). We conduct two additional synthetic cases studies. The first synthetic case is an idealized scenario

where diffusion is limited in 1.5 Ma ice ("Low diffusion" in Table 1). The second case aims to simulate a scenario with higher diffusive smoothing ("High diffusion in Table 1). Figure 4 shows how the SDR of CO<sub>2</sub> and the O<sub>2</sub>/N<sub>2</sub> ratio evolve for each model run. The simulation of CO<sub>2</sub> smoothing at EDC yields an SDR of <0.1 % and an O<sub>2</sub>/N<sub>2</sub> SDR of 3 % at 0.8 Ma, the ice core record limit. The simulation of CO<sub>2</sub> smoothing at Dome F yields an SDR of <0.1 % and an O<sub>2</sub>/N<sub>2</sub> SDR of 2.7 % at 0.75 Ma, the age limit in the model run. Both the low and high diffusion model runs also produce SDRs for each gas below 10 % at 0.8 Ma, indicating that gas signal diffusion within the age range of the current ice core record is minimal. However, the SDR values between the low and high diffusion model runs diverge after 0.8 Ma, with the high smoothing simulation yielding an SDR of 84 % for O<sub>2</sub>/N<sub>2</sub> at 1.5 Ma and the low smoothing simulation yielding an SDR of 45 % at the same age. While atmospheric signals are well preserved in the ice sheet to 0.8 Ma, they may be significantly smoothed by 1.5 Ma. The simulated smoothing in ice older than 0.8 Ma is driven by extreme layer thinning and warmer temperatures near the bed. It will be important to consider gas diffusion in older ice despite the lack of diffusion in the current record as orbital scale variations in gases in the current ice core record cannot be used to estimate diffusion.


|                | Acc. rate (cm yr <sup>-</sup> 1) | Ice thickness (m) | Surface temp (°C) | GHF (mW m <sup>-2</sup> ) | p   |
|----------------|----------------------------------|-------------------|-------------------|---------------------------|-----|
| EDC            | 1.82                             | 3153              | -60.95            | 53.3                      | 3.8 |
| DF             | 2.1                              | 3032              | -58               | 55                        | 4   |
| Low diffusion  | 2                                | 3000              | -60               | 40                        | 4   |
| High diffusion | 4                                | 2700              | -60               | 40                        | 4   |

Table 1: Forcing values used to model each scenario to assess how reliable estimates of diffusive smoothing from existing ice core gas concentration measurements are. EDC surface temperatures approximate a glacial-cycle average. The SDR from 0 to 1.5 Ma of each scenario is shown in Fig. 4.

Figure 4: SDR for CO<sub>2</sub> (dashed) and the O<sub>2</sub>/N<sub>2</sub> ratio (dotted) modeled with EPICA Dome C forcings (green), Dome F forcings (magenta), a low diffusion scenario (blue), and a high diffusion scenario (red). The inset shows each case between 0.4 and 0.8 Ma.

## 3.2 SDR sensitivity to input parameters




Here, we investigate the sensitivity of gas diffusion to four input parameters to the models: accumulation rate, ice thickness, surface temperature, and GHF. We hold three parameters constant and vary the fourth to gain an intuition for how SDR responds to each forcing parameter separately. Parameter values are chosen to keep the melt rate at zero. This ensures that the oldest ages are represented in this analysis and not destroyed by basal melt. The parameter values also are different between the CO<sub>2</sub> and the O<sub>2</sub>/N<sub>2</sub> ratio model runs. We consider separate sets of parameters for CO<sub>2</sub> and O<sub>2</sub>/N<sub>2</sub> because of the large difference in the diffusion rates of CO<sub>2</sub> and the O<sub>2</sub>/N<sub>2</sub> ratio. Fixed parameter values for each gas are shown in Table 2. The range for each variable parameter is based on the physical properties of the study area. The effects of the parameters are interrelated. For example, the effect of varying accumulation depends on the assumed temperature, ice sheet thickness, and GHF. So, this analysis should be considered qualitative. Figures 5 and 6 show CO<sub>2</sub> and the O<sub>2</sub>/N<sub>2</sub> ratio SDR sensitivity to input parameters.

|                 | Acc. rate (cm yr <sup>-1</sup> ) | Ice thickness (m) | Surface temp. (°C) | GHF (mW m <sup>-2</sup> ) |
|-----------------|----------------------------------|-------------------|--------------------|---------------------------|
| CO <sub>2</sub> | 5                                | 2800              | -60                | 50                        |
| $O_2/N_2$       | 2                                | 2800              | -65                | 40                        |

Table 2: Fixed parameter values for sensitivity analysis. We use different parameter values for each gas due to their different levels of diffusive smoothing. Values are set to maintain zero melt rate.

The SDR of CO<sub>2</sub> increases roughly linearly with accumulation rate, surface temperature, and GHF, and decreases with ice thickness. Increasing accumulation yields increased SDR (Figure 5a). Higher accumulation enhances signal smoothing because the thinner layers near the bed cause steeper gas concentration gradients. The steep gradients increase diffusion more than the decrease in ice temperature from greater vertical advection of cold surface temperatures. Additionally, the relatively steep slope in Fig. 5a shows that accumulation rate has the largest impact on CO<sub>2</sub> signal preservation. The effect of surface temperature and GHF are simpler. Lower values reduce SDR because of colder temperatures and the changes are near linear for both (Figure 5c and d). The response to ice thickness is more complex. Thicker ice columns result in lower SDR due to thicker layers and shallower gas concentration gradients. However, ice columns larger than ~3000 m have a greater likelihood of basal melting, potentially removing old ice. Therefore, the sensitivity in Fig. 5b should not be interpreted to suggest maximizing ice thickness will minimize SDR. We explore scenarios with basal melt in more detail in Fig. 7.

Figure 5: Sensitivity of CO<sub>2</sub> SDR to (a) accumulation rate, (b) ice thickness, (c) surface temperature, and (d) GHF. Parameter values are chosen to keep the melt rate at zero.

The O<sub>2</sub>/N<sub>2</sub> ratio model runs show similar results to those for CO<sub>2</sub>. Accumulation rate's impact on gas concentration gradients dominates the impact from colder temperatures (Figure 6a), and its relatively steep slope also shows its importance among the forcings. The O<sub>2</sub>/N<sub>2</sub> ratio SDR is mildly more sensitive to GHF, due to higher sensitivity of O<sub>2</sub> permeability to temperature compared to CO<sub>2</sub>. Similarly to CO<sub>2</sub>, minimizing surface temperature and GHF will minimize SDR for the O<sub>2</sub>/N<sub>2</sub> ratio, too (Figure 6c and d). Unlike CO<sub>2</sub>, ice thickness has a relatively weak effect on the O<sub>2</sub>/N<sub>2</sub> ratio SDR. In thicker ice columns, we find the impact on temperature slightly dominates the impact on gas concentration gradients as demonstrated by the slightly

positive slope in Fig. 6b. This arises from a higher sensitivity to temperature in the permeability of  $O_2$  compared to  $CO_2$ . However, in shallower ice columns, where temperatures are cooler compared to thicker ice columns, the relatively flat slope implies the impacts on temperature and layer thinning are about equal and opposite. The lower dissociation pressure of  $O_2$  compared to  $CO_2$  causes more diffusion in the warmer temperatures of thick ice columns. Ice thickness's weak effect on the  $O_2/N_2$  ratio SDR is further highlighted in Fig. 7.

Figure 6: Sensitivity of O<sub>2</sub>/N<sub>2</sub> ratio SDR to (a) accumulation rate, (b) ice thickness, (c) surface temperature, and (d) GHF. Parameter values are chosen to keep the melt rate at zero.

Basal melt is important to consider for SDR and site selection. Under scenarios where basal melt rates are low, about 0.1 mm yr<sup>-1</sup>, ice older than 1.5 Ma is removed but ice of 1.5 Ma age remains. The melting of the older ice, without removing the 1.5 Ma ice, creates additional vertical space for the 1.5 Ma ice to occupy, thus reducing the thinning. The reduced thinning of 1.5 Ma ice thus results in improving gas signal preservation. Figure 7 shows SDR values for CO<sub>2</sub> and the O<sub>2</sub>/N<sub>2</sub> ratio in 1.5 Ma ice under varying GHF and ice thickness, with fixed accumulation rates of 3 cm yr<sup>-1</sup> and surface temperatures of -60°C. The signal preservation enhancing effect of low melt rates is apparent at the boundary between preserved 1.5 Ma ice and melted ice, where SDR values are lower by about 5 % (Figure 7). This is most important for CO<sub>2</sub> signal preservation, as we see that the O<sub>2</sub>/N<sub>2</sub> signals are almost fully damped above 45 mW m<sup>-2</sup>. However, producing such a low melt rate requires a specific range of forcing values in this simplified model, so a site with these conditions is likely rare.

Figure 7: SDR in 1.5 Ma ice for  $CO_2$  (left) and the  $O_2/N_2$  ratio (right) with varying GHF and ice thickness. Note the different color scales. Accumulation rate is fixed at 3 cm yr<sup>-1</sup>; surface temperature is fixed at -60 °C.

#### 3.3 Atmospheric gas diffusion in the interior of the east Antarctic ice sheet





The gas diffusion model is now applied to the COLDEX study area (Figure 8). We will first discuss the distribution of old ice in each of the three regions – the Saddle Region, the Deep Basin, and the Foothills – then discuss the CO<sub>2</sub> and the O<sub>2</sub>/N<sub>2</sub> ratio SDRs. The accumulation rate and ice thickness inferred from the airborne radar data (Young et al., 2025) is shown in Fig. 8a and b and the surface temperature forcing in Fig. 8c. Estimates of GHF (Stål et al., 2020; Shen et al., 2020; An et al., 2015; Fox-Maule et al., 2005; Martos et al., 2017; Shapiro and Ritzwoller, 2004; Purucker, 2013; Hazzard et al., 2024) show considerable variability and their relative accuracy cannot be assessed; therefore, we use uniform GHF values of 45, 50, and 55 mW m<sup>-2</sup>. We consider the near-basal ice age as the age of ice that is 2 % of the ice column's thickness above the bed (Figure 8g–i), which is between 40 m and 80 m above the bed in the COLDEX study area. We use the 2 % threshold to facilitate comparison among the study area.

The Saddle Region and the Foothills retain old ice, while the Deep Basin has mostly lost the oldest ice to basal melt (Figure 8g–i). Only in the low GHF scenario does the Deep Basin retain old ice. Figure 8d-f shows how each GHF scenario increases the basal melt, particularly in the Deep Basin. It is also important to note that the Saddle Region is downstream from the Deep Basin and advection is not considered in our model; we discuss the effects of advection in Sect. 4.1. The Foothills contain the oldest ice under all GHF scenarios, but the calculated age is variable due to the considerable variability in ice thickness in the region.

We discuss SDR in terms of its value at 1.5 Ma. The lowest CO<sub>2</sub> SDR (Figure 8j–l) is in the Foothills and in the Deep Basin when there is old ice present. CO<sub>2</sub> SDR tends to be higher in the Saddle Region where accumulation rates are higher, consistent with the results in Sect. 3.2. The variable ice thickness in the Foothills contributes to variability in CO<sub>2</sub> SDR but is not as

significant as the variability in ice age. For each GHF scenario, the CO<sub>2</sub> SDR varies marginally in the Foothills (~3–10 %) and varies more significantly in the Saddle Region (~18–30 %).

The O<sub>2</sub>/N<sub>2</sub> ratio SDR (Figure 8m–o) follows a similar pattern to the CO<sub>2</sub> SDR. The lowest O<sub>2</sub>/N<sub>2</sub> ratio SDR is in the Foothills, although it is still much higher than the SDR for CO<sub>2</sub>. Both the Deep Basin and Saddle Region have nearly 100 % SDR. In the Deep Basin, this is because of the warm temperatures in the deep ice while in the Saddle Region it is because of the high accumulation rate and steep gas concentration gradients in the basal ice. The Foothills is the only region with SDR significantly below 1, due to the effects of low melt rates as described in Sect. 3.2. Although, there is considerable variability in the SDR in the Foothills.




Uncertainties in GHF have a larger effect on the distribution of old ice compared to the diffusion of gas signals. In the Deep Basin, where ice thickness is consistently around or above 3000 m, we see the amount of old ice present changes dramatically with different GHF values. At 45 mW m<sup>-2</sup>, old ice is abundant in the basin, only absent where ice thickness nears 4000 m. However, at 50 mW m<sup>-2</sup> most of the old ice is gone except towards grid-south, and at 55 mW m<sup>-2</sup> there is virtually no old ice in the basin. Assuming uniform GHF, the oldest ice will be located near either the South Pole, the Foothills, or both according to the model, with the Foothills having better conditions for atmospheric gas signal preservation. Additionally, advection will affect the presence of old ice in the Saddle Region, as discussed in Sect. 4.1.

Figure 8: (a) Accumulation rate and (b) ice thickness were measured or inferred with aerial radar, provided by COLDEX. (c) Surface temperature is interpolated over the region based on the Easting from the South Pole. Each model output is calculated under three GHF scenarios: 45, 50, and 55 mW m<sup>-2</sup>. (d–e) Basal melt rate and (g–i) the age of near-basal ice (2 % of ice thickness above the bed,  $\sim$ 20–40 m) as calculated by the 1D, steady-state ice and heat flow model. (j–l) SDR for CO<sub>2</sub> in 1.5 Ma ice; grey points indicate where 1.5 Ma ice is melted. (m–o) SDR for the O<sub>2</sub>/N<sub>2</sub> ratio in 1.5 Ma. The yellow star represents the South Pole.

## 4 Discussion





## 4.1 Impact of Advection

We have presented 1D results and have not assessed the impact of advection. Advection will be important to consider because even the slow horizontal velocities in the region, 1-2 m yr<sup>-1</sup> in the survey region, imply that 1.5 Ma ice has travelled hundreds of kilometers. Unfortunately, modeling SDR for the full study area with flowline models is intractable both because of the computational time and the increasing number of assumptions needed to force the model. We note that the gas diffusion model can be applied to any ice thinning and temperature scenario and thus is compatible with output from ice flow models as long as this information is tracked (e.g. Chung et al., 2024; Parrenin et al., 2025). Consideration of specific ice core sites should employ flow band models where parameter ranges for the upstream area can be carefully considered.

The 1D model results are best for gaining qualitative information about the broader regions which are likely to best preserve atmospheric gas records. The primary limitation is that small scale variations in forcing parameters, such as bedrock topography, can produce results that appear favorable but where advection would have instead significantly imprinted the conditions from upstream on the ice at that location. Thus, considering the conditions in the upstream direction is useful for interpreting the results at any individual location. For example, in our results the model signifies the presence of old ice in the Saddle region (Figure 8g–i). However, ice flows from the Foothills, through the Deep Basin, where old ice likely melts away, before reaching the Saddle Region. Thus, the Saddle Region is less likely to preserve old ice than the 1D model results may suggest.

## 4.2 Implications for ice core site selection

The most favorable site conditions for the preservation of gas signals in a deep ice core are where accumulation rates, basal melt rates, and surface temperatures are low, and ice thickness is moderate, between ~2500 and ~3000 m. If a location with low GHF can be identified, then greater ice thicknesses are preferred because basal melt is avoided and the gas concentration gradients are smaller, reducing the amount of diffusion.

The Deep Basin, due to its large ice thickness, results in too much basal melt, removing the old ice. Even in the low GHF scenario (45 mW m<sup>-2</sup>), some of this region experiences basal melt (Figure 8d). We note that if the GHF is unusually low, the Deep Basin could provide exceptional preservation conditions; however, geophysical observations supporting such a unique combination would be necessary to justify further exploration in this region. In the Saddle Region, accumulation rates are relatively high (~6 cm yr<sup>-1</sup>), steepening gas concentration gradients in the basal ice and increasing signal diffusion. The SDR is higher in the Saddle Region than in the Foothills, as seen in Fig. 9. We also note that the ice has flowed across the Deep Basin, so if melt has removed the old ice in the Deep Basin, it would not be present in the Saddle Region. Thus, both the Deep Basin and the Saddle Region are not optimal locations for a deep ice core. While there remains the possibility of finding an ice core site in these regions, it would likely require an unusually low GHF in the Deep Basin (e.g. Hazzard et al., 2024) so that ice is not removed there.

The lower accumulation rates and moderate ice thickness make the Foothills a more promising region for a deep ice core. Old ice is more common and SDRs are lowest in this region. Within the Foothills, the grid-south portion of the foothills have thicker ice columns than the grid-north portion, increasing melt rates and decreasing the old ice present. This leaves the gridnorth foothills as the most promising location (Figure 9). Even though this is the most promising region, the presence of 1.5 Ma ice is variable and should be evaluated at a particular site to ensure its presence. Sites where the melt rate might be low 400 enough, but non-zero, to decrease SDR as described in Sect. 3.2 should not be sought, since the narrow range of necessary forcings, the uncertainties associated with the model, and the impact of advection make looking for such a site impractical. Our results show that CO<sub>2</sub> SDR for 1.5 Ma ice does not exceed 13 % in the grid-north Foothills and averages 5 % given 50 mW m<sup>-2</sup> GHF (Figure 8k). However, the SDR for the O<sub>2</sub>/N<sub>2</sub> ratio in the grid-north Foothills is still high, with a mean value of 89 % (Figure 81). There is uncertainty associated with the fraction of gas signal lost to diffusion in the model arising from 405 uncertain gas parameter values (see Sect. 4.4). Nonetheless, we still expect that CO<sub>2</sub> signals will experience only a small amount of diffusion, while the O<sub>2</sub>/N<sub>2</sub> ratio will experience significantly more. The difference between the SDR for CO<sub>2</sub> and the O<sub>2</sub>/N<sub>2</sub> ratio is partly due to the different frequency in the climate record; the SDR for a 20 kyr CO<sub>2</sub> signal is 28 % for a typical site in the Foothills whereas the 40 kyr SDR is 8 %. Therefore, even though old ice is likely present in the grid-north Foothills and CO<sub>2</sub> may be well preserved, the damping of precession signals in the O<sub>2</sub>/N<sub>2</sub> ratio from this region may preclude 410 dating of the ice core. Measurements of total air content (TAC) may provide another method for dating an old ice core (e.g. Raynuad et al., 2007), but this method comes with its own complications (Vudayagiri et al., 2025) as climate and elevation changes can overprint the insolation signal. Future work could consider how TAC signals may diffuse in old ice. For 1.2 Ma ice (when the MPT is thought to have begun; Clark et al., 2006) the SDR for the O<sub>2</sub>/N<sub>2</sub> ratio is lower than at 1.5 Ma (~0.5–0.7 in the grid-north Foothills), so it may be possible to track precession cycles through the MPT.

Figure 9: CO<sub>2</sub> SDR in 1.5 Ma ice under 50 mW m<sup>-2</sup> GHF (Figure 8k) and ice thickness and elevation (Figure 2a).

## 4.3 Potential impacts of basal ice units

The 1D model we use assumes a continuous depth-age relation to the bed; however, radar imaging shows that stratigraphic layering is lost hundreds of meters above the bed and finds diffuse scattering in portions of the deep ice (Young et al., 2025). Work around Dome C suggests there may be basal layers of stagnant ice (Lilien et al., 2021; Chung et al., 2023; 2024), which can affect the inferred depth-age relationship. Analysis of the radar data at ice core locations (Mutter and Holschuh, 2025) show that the continuous climate record can be preserved at depths with diffuse scattering. We assume that the climate record is undisturbed to the bed in this study. However, if instead the basal unit is mechanically decoupled (e.g. Lilien et al., 2021; Chung et al., 2024), then the thickness of the deforming ice column is less than the total ice thickness. This would result in greater layer thinning as well as colder temperatures because the old ice is higher in the ice column.

|                             | Deforming ice | Basal ice layer | Total ice     | GHF (mW m <sup>-2</sup> ) | CO <sub>2</sub> SDR | O <sub>2</sub> /N <sub>2</sub> SDR |
|-----------------------------|---------------|-----------------|---------------|---------------------------|---------------------|------------------------------------|
|                             | thickness (m) | thickness (m)   | thickness (m) |                           | (1.5 Ma)            | (1.5 Ma)                           |
| Case 1 (no basal ice)       | 2700          | 0               | 2700          | 50                        | 5.7 %               | 92.0 %                             |
| Case 2                      | 2430          | 270             | 2700          | 50                        | 6.3 %               | 88.1 %                             |
| (layer thinning effect)     | 2130          | 270             | 2700          | 30                        | 0.5 70              | 00.1 70                            |
| Case 3 (temperature effect) | 2700          | 300             | 3000          | 49.2                      | 5.4 %               | 89.6 %                             |

Table 3: Deforming ice thickness, basal ice layer thickness, total ice thickness, GHF, and SDR at 1.5 Ma for each basal ice layer case.

Case 1 is a control which simulates a 2700 m deforming ice column. Case 2 simulates the presence of a 270 m basal ice layer to assess the impact of increased layer thinning from a basal ice layer. Case 3 simulates the presence of a 300 m basal ice layer to assess the impact of decreased temperature from a basal ice layer. Each case uses an accumulation rate of 2 cm yr<sup>-1</sup> and a surface temperature of -60 °C.




Figure 10: The temperature (left) and age (right) profiles for each model case. The age profiles of Cases 1 and 3 are the same. Note that depth is measured from the surface, not as height above the bed.

To assess the impact of a potential basal ice layer, we perform three model runs and compare their SDRs. Case 1 is the control with 2700 m ice thickness, 50 mW m<sup>-2</sup> GHF, 2 cm yr<sup>-1</sup> accumulation rate, and -60 °C surface temperature. Case 2 investigates the impact of a change in layer thickness while keeping the temperature the same. Case 3 investigates the impact of a change in temperature while keeping the layer thickness the same.

The temperature and age profiles of the three cases are shown in Fig. 10. We do not directly incorporate a non-deforming basal layer and instead adjust the GHF to simulate its impacts. Case 2 simulates a 10 % (270 m) basal ice layer to assess the impact of layer thinning from a non-deforming basal ice layer; we model a thinner ice column (2430 m) but match the Case 1 temperature. Case 3 simulates a 3000 m ice column with the bottom 10 % (300 m) non-deforming to assess the impact of a change in ice temperature from a basal ice layer; we model the same thickness (2700 m, so the depth-age profile is the same), but match the temperature profile to a model run of Case 1 extended to 3000 m ice thickness (Case 1\* in Fig. 10) by reducing the GHF. The cases and resulting SDR are summarized in Table 3.

In Case 2, a thinner deforming ice column increases CO<sub>2</sub> SDR and decreases the O<sub>2</sub>/N<sub>2</sub> ratio SDR. This aligns with what we would expect given the parameter sensitivity analysis in Sect. 3.2 (see Fig. 5b and Fig. 6b). In Case 3, a lower ice temperature from the presence of a basal ice layer decreases the CO<sub>2</sub> and the O<sub>2</sub>/N<sub>2</sub> ratio SDRs. This also aligns with our expectations since the presence of a basal ice layer decreases the ice temperature, reducing diffusion. The difference in SDR for all cases is small,

so we don't expect the presence of a basal ice layer to dramatically alter the preservation of gas signals. In each case,  $CO_2$  remains well preserved and the  $O_2/N_2$  ratio is poorly preserved.

## 4.4 Uncertainties from gas parameters






We perform sensitivity studies to assess the possible implications of the uncertainty in gas permeation parameters for our result. All sensitivity runs use forcing values identical to those in Fig. 3 (2 cm yr<sup>-1</sup> accumulation rate, 3000 m ice thickness, 50 mW m<sup>-2</sup> GHF, -60 °C surface temperature). Salamatin et al. (2001) report a ±15 % uncertainty for the permeation coefficient of O<sub>2</sub>, so we conduct simulations using permeation coefficients that span this range. Ahn et al. (2008) do not provide an uncertainty for the permeability of CO<sub>2</sub> so we apply the same 15 % uncertainty range. Oyabu et al. (2021) have previously demonstrated that O<sub>2</sub> permeation values an order of magnitude faster than those we consider here yield unrealistically high smoothing compared to O<sub>2</sub>/N<sub>2</sub> measurements from the Dome Fuji core. Those researchers also show that the permeation coefficients of Salamatin et al. (2001) yield diffusive smoothing in reasonable agreement with the Dome Fuji measurements, increasing confidence in our parameterizations. However, the permeation values provided by Salamatin et al. (2001) (the "slow set") are tested over a limited temperature range, and the values given by Ikeda-Fukazawa et al. (2004) (the "fast set") approach the values in the "slow set" at higher temperatures, like those near the bed of the ice sheet. Thus, our use of the "slow set" in this study may provide a more conservative estimation of diffusive smoothing of O<sub>2</sub>/N<sub>2</sub>. Future ice core measurements may improve our understanding of the temperature dependence of these permeation coefficients. Table 4 shows the SDR for each gas from each model run. Uncertainty in the permeation coefficient of CO<sub>2</sub> affects the SDR by <1 %. Uncertainty in the permeation coefficient of  $O_2$  can change the  $O_2/N_2$  ratio SDR by about  $\pm 5$  %. The study area will be affected by this uncertainty equally, and so the pattern of where SDR is low will not be altered significantly by these results. The expected SDR we find in this study is much more dependent on ice sheet physical properties than the uncertainty in gas permeation.

|                                             | Control | Permeation coefficient +15% (-15%) |
|---------------------------------------------|---------|------------------------------------|
| CO <sub>2</sub> SDR (1.5 Ma)                | 2.8 %   | 3.3 % (2.4 %)                      |
| O <sub>2</sub> /N <sub>2</sub> SDR (1.5 Ma) | 80.4 %  | 84.7 % (75.0 %)                    |

Table 4: CO<sub>2</sub> and O<sub>2</sub>/N<sub>2</sub> ratio SDR based on permeation coefficient uncertainties. Uncertainty ranges are based on the uncertainty of the permeation coefficient of O<sub>2</sub> (Salamatin et al., 2001). All model runs are done with the input forcings shown in Fig. 3. "Control" refers to the model run with no alterations to gas parameter values.

#### 480 5 Conclusions

Gas signal preservation in a 1.5 Ma ice core from the COLDEX study area between South Pole and Dome A was investigated using gas diffusion models. We expand on the work of Bereiter et al. (2014) who showed that CO<sub>2</sub> variations, but not the O<sub>2</sub>/N<sub>2</sub>

ratio, are likely preserved at an idealized location similar to the Little Dome C site that the Beyond EPICA Oldest Ice program has recently completed drilling. We find that the conditions most likely to preserve atmospheric gas signals are a low accumulation rate, minimal surface temperature, minimal GHF, and moderate ice thickness between 2500 and 3000 m. We identify the Foothills region, the portion of the study area closest to Dome A, as more likely to preserve atmospheric gas records. CO<sub>2</sub> signals (with a period of 40 kyr) do not lose more than 13 % of their amplitude in the Foothills compared to up to 43 % elsewhere in the study area. The Foothills also better preserve the O<sub>2</sub>/N<sub>2</sub> ratio; however, O<sub>2</sub>/N<sub>2</sub> signal preservation is poor with 89 % signal loss on average in the Foothills. The grid north portion of the Foothills has more moderate ice thickness and is less likely than the grid south portion to lose old ice to basal melt.

Our analysis focused on four boundary conditions. The accumulation rate is the dominant factor for diffusion: high accumulation rates lead to more layer thinning in old ice, resulting in an increased gas concentration gradient and thus more diffusion. This outweighs any reduction in diffusion from colder temperatures in the ice column. Thicker ice has a similar effect to low accumulation rates by reducing layer thinning. For CO<sub>2</sub>, this results in better gas preservation; the O<sub>2</sub>/N<sub>2</sub> ratio is not as sensitive to this due to a greater effect from higher temperatures in thicker ice columns. However, thicker ice also promotes basal melting and removal of 1.5 Ma ice. Thus, moderate ice thickness of 2500 m to 3000 m is better with the important caveat that thicker ice is better for CO<sub>2</sub> preservation if the GHF is low enough to not induce substantial basal melt. GHF is the least constrained parameter, introducing large uncertainty in whether 1.5 Ma ice will be preserved. Atmospheric gas preservation also has uncertainty due to unknown gas parameter coefficients. We note that gas parameter coefficients will affect all areas of the study area similarly, so while the absolute preservation may be uncertain, the relative pattern of preservation should be consistent. The Foothills region, particularly to the grid north, is the most likely region to preserve atmospheric gas records reaching 1.5 Ma and should be explored in more detail to determine whether it is suitable for a deep ice core.

Our results suggest that the recovery of a continuous ice core record back to 1.5 Ma would not experience significant diffusive smoothing of  $CO_2$  signals, under the recommended conditions. Thus, such an ice core will provide insights into the role  $CO_2$  played in the MPT. However, dating such an ice core will likely require a method other than precession cycles apparent in the  $O_2/N_2$  ratio. The  $O_2/N_2$  ratio diffuses significantly in ice older than 1 Ma, but if high resolution measurements can distinguish precession cycles in a damped  $O_2/N_2$  record, some dating may still be achievable.

## 510 Code Availability






Model code is available at https://doi.org/10.5281/zenodo.15347004. We utilized the following MATLAB packages in creating figures: Red Blue Colormap (Adam, 2023); Scientific Colour Maps (Crameri, 2019); Antarctic Mapping Tools for MATLAB (Greene et al., 2017); The Climate Data Toolbox for MATLAB (Greene et al., 2019); Antarctic Boundaries for IPY 2007-2009 from Satellite Radar, Version 2 (Mouginot et al., 2017).

#### **Author Contribution**

MJS and TJF designed the temperature and age-depth model and conducted experimentation and analysis. JDP designed the gas diffusion model. SY and DAY created figure 2. SY created figure 9. DAY, SS, DB, and MK contributed to collecting accumulation rate and ice thickness data. All authors contributed to editing the manuscript.

# 520

### **Competing Interests**

The authors declare that they have no conflict of interest.

## Acknowledgements

This work was supported by the U.S. National Science Foundation Center for Oldest Ice Exploration (NSF COLDEX), an NSF Science and Technology Center (NSF 2019719) and U.S. National Science Foundation Antarctic Glaciology program (NSF 1851022). Support for M.J. Sailer was provided by National Oceanographic and Atmospheric Administration Cooperative Institute for Climate, Ocean, & Ecosystem Studies and NSF COLDEX Research Experience for Undergraduates as well at the University of Washington Opportunities in Glacier InVEstigation. We thank the NSF Office of Polar Programs, the NSF Office of Integrative Activities, Oregon State University, and University of Washington for financial and infrastructure support, and the NSF Antarctic Infrastructure and Logistics Program and the Antarctic Support Contractor for logistical support. We thank Kenn Borek Air and South Pole Station personnel for their support.

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
