# Peer review of "Ice core site considerations from modeling $CO_2$ and $O_2/N_2$ ratio diffusion in interior East Antarctica"

_EGUsphere, 2025_

## Author Comment (AC1)

Thank you to the reviewer for providing specific and valuable feedback on our manuscript. We have carefully reviewed and incorporated the recommendations into a revised manuscript and describe the changes in the following response. The reviewer's responses are written in black text, and our answers are written in red text. Revised sentences in the manuscript are written in blue text.
* * *
**Review of Sailer et al.: Ice core site considerations from modeling CO2 and O2/N2 ratio diffusion in interior East Antarctica**

This paper presents simulations of how much CO2 and O2/N2 signals may be attenuated by gas diffusion within ice sheets preserving 1.5 million years of climate history. Also, using both an ice and heat flow model and a gas diffusion model, the authors predict regions across a broad area from the South Pole to Dome A where million-year-old ice is likely to preserve atmospheric signals with higher amplitudes. This study provides valuable information for selecting drilling sites for the NSF COLDEX project. Moreover, their estimates could be validated in the future through ongoing oldest ice core projects, which could in turn help constrain the diffusion coefficients of gas molecules in ice—parameters that are otherwise extremely difficult to measure.

Overall, I consider this paper suitable for publication in Climate of the Past after minor revisions. However, I recommend that all the points raised in this review be carefully addressed before the manuscript is accepted.

We thank the referee for their helpful comments. We have incorporated the suggested references and describe our responses in detail below.

**General comments:**

Gas diffusion in ice is controlled by both the gas concentration gradient and temperature. This paper presents a set of sensitivity experiments by varying parameters such as accumulation rate, ice thickness, surface temperature, and geothermal heat flux (GHF), and it is a nice aspect of the work that the authors investigated how each of these

parameters affects CO2 and O2 diffusion in the ice. However, to facilitate more intuitive understanding of the results, I suggest the authors include the vertical temperature profiles corresponding to the ice thicknesses used in each experiment. This would also help the reader better interpret the results of the experiments that assume the presence of stagnant ice near the bed.

We think the referee is most interested in section 4.3, the discussion of the basal ice unit. We agree that the discussion was difficult to follow, and we have created a figure to illustrate the different cases (below). The figure shows both the temperature-depth profiles and age-depth profiles. We have revised the text of section 4.3, which is included in the lini-by-line responses below.

We also note that we show an example vertical temperature profile in Figure 3 and updated this figure with two additional subpanels to help show the temperature that packets of ice of different ages are experiencing and how thin the layers have become. We hope this will help give readers better intuition for the evolution of the conditions that drive diffusion.

The comment may also pertain to Figure 5 and 6 where we do the uncertainty analysis, but we can't figure out a way to succinctly show the vertical temperature profiles that vary with ice thickness, and why to focus only on ice thickness. We therefore do not make any changes to this section.

[Figure]

Figure 10: The temperature (left) and age (right) profiles for each model case. The age profiles of Cases 1 and 3 are the same. Note that depth is measured from the surface, not as height above the bed.

Regarding the captions of figures and tables: in many cases (notably Fig. 4, Fig. 5, Fig. 6, Fig. 7, Fig. 8, Fig. 9, Table 3, and Table 4), the first sentence is descriptive. It would be more

appropriate to begin each caption by stating clearly what the figure or table is showing. Moreover, several captions include explanations of the methodology or interpretation of the results, which should instead be included in the main text. I recommend that the authors review and revise all figure and table captions accordingly.

We have made various changes to these captions to reflect this comment:

Figure 4: SDR for $CO_2$ (dashed) and the $O_2/N_2$ ratio (dotted) modeled with EPICA Dome C forcings (green), Dome F forcings (magenta), a low diffusion scenario (blue), and a high diffusion scenario (red). The inset shows each case between 0.4 and 0.8 Ma.

Figure 5: Sensitivity of $CO_2$ SDR to (a) accumulation rate, (b) ice thickness, (c) surface temperature, and (d) GHF. Parameter values are chosen to keep the melt rate at zero.

Figure 6: Sensitivity of $O_2/N_2$ ratio SDR to (a) accumulation rate, (b) ice thickness, (c) surface temperature, and (d) GHF. Parameter values are chosen to keep the melt rate at zero.

Figure 7: SDR in 1.5 Ma ice for $CO_2$ (left) and the $O_2/N_2$ ratio (right) with varying GHF and ice thickness. Note the different color scales. Accumulation rate is fixed at 3 cm yr-1; surface temperature is fixed at -60 ℃.

Figure 8: (a) Accumulation rate and (b) ice thickness were measured or inferred with aerial radar, provided by COLDEX. (c) Surface temperature is interpolated over the region based on the Easting from the South Pole. Each model output is calculated under three GHF scenarios: 45, 50, and 55 mW m-2. (d–e) Basal melt rate and (g–i) the age of near-basal ice (2 % of ice thickness above the bed, ~20–40 m) as calculated by the 1D, steady-state ice and heat flow model. (j–l) SDR for $CO_2$ in 1.5 Ma ice; grey points indicate where 1.5 Ma ice is melted. (m–o) SDR for the $O_2/N_2$ ratio in 1.5 Ma. The yellow star represents the South Pole.

Figure 9: $CO_2$ SDR in 1.5 Ma ice under 50 mW m-2 GHF (Fig. 8k) and ice thickness.

Table 3: Deforming ice thickness, basal ice layer thickness, total ice thickness, GHF, and SDR at 1.5 Ma for each basal ice layer case. Case 1 is a control which simulates a 2700 m deforming ice column. Case 2 simulates the presence of a 270 m basal ice layer to assess the impact of increased layer thinning from a basal ice layer. Case 3 simulates the presence of a 300 m basal ice layer to assess the impact of decreased temperature from a basal ice layer. Each case uses an accumulation rate of 2 cm yr-1 and a surface temperature of -60 ℃.

Table 4: $CO_2$ and $O_2/N_2$ ratio SDR based on permeation coefficient uncertainties. Uncertainty ranges based on the uncertainty of the permeation coefficient of $O_2$

(Salamatin et al., 2001). All model runs are done with the input forcings shown in Fig. 3. "Control" refers to the model run with no alterations to gas parameter values.

There are some citations to works that are in preparation, submitted, or in review. In general, such works should not be cited, as it is not possible to verify whether the citation or the associated discussion is appropriate. In particular, citing works in preparation is highly inappropriate. While I leave the final decision to the Editor, I believe that only works that are publicly accessible should be cited.

We are aware of the unpublished nature of these works and will ensure that a publicly accessible paper or dataset is available in the revised version of the manuscript.

Line 49 and more: Young et al., submitted

Now in press

Line 96: Singh et al., in prep

Line 360: Fudge et al., in preparation

Line 364: Parrenin et al., in review (there is no info in the reference list)

We have added the reference: Parrenin, F., Chung, A., and Martín, C.: age_flow_line-1.0: a fast and accurate numerical age model for a pseudo-steady flow tube of an ice sheet, EGUsphere [preprint], https://doi.org/10.5194/egusphere-2024-3411, 2025.

**More specific comments**

Lines 60 and 63: Line breaks are unnecessary here.

Agree, we have made the recommended change.

Line 69: The sentence "Feedbacks with the decreased…" is unclear. Please clarify the meaning and provide a proper citation.

We have removed this sentence as we believe it was confusing and unnecessary to include.

Line 70: The reference to Stolper et al. (2016) is inappropriate here. It is related to reconstructing atmospheric O2 histories, not to dating. You should instead cite Kawamura et al. (2007, https://doi.org/10.1038/nature06015), who first used O2/N2 for dating. The correct reference for AICC2023 is Bouchet et al. (2023).

We have added the recommended citation and corrected the reference for AICC2023.

Line 72: Please consider adding citations to Kawamura et al. (2007) and Fujita et al. (2009, https://doi.org/10.1029/2008JF001143) here. They also discuss the mechanism for the relationship between O2/N2 and local summer insolation.

We have added the recommended citations.

Line 73: The phrase "this age" is unclear. In addition, a citation is needed to support the statement that "This is the most reliable method." For example, Oyabu et al. (2022, https://doi.org/10.1016/j.quascirev.2022.107754) provided the first evidence that O2/N2-based chronology is highly reliable, by demonstrating that O2/N2-derived ages show no phase lag relative to insolation cycles within the estimated uncertainties, based on comparison with U–Th-dated speleothem δ18O.

We have added the recommended reference and changed the wording of the text to be less unequivocal:

This is a key method in dating gases in the oldest ice cores (Oyabu et al., 2022).

Lines 90 - 95 and Figure 2: The colors in Figure 2 make it difficult to distinguish between 2000, 2500, and 3000 m cases. Also, since the spatial extent is unclear, I suggest including a map of the entire Antarctic continent to show the location of the zoomed-in region. Please also plot the location of Dome A.

We have changed the color scale and added a map for reference:

[Figure]

Line 97: What accumulation rate value is used for the past 4.7 ka?

The accumulation rates we used in the study region are inferred and scaled based on an englacial layer dated to 4.7 ka. We have changed the text to be more clear:

The accumulation rates across the study area are inferred from an englacial layer dated to 4.7 ka (Singh et al., in prep) and given in ice equivalent. Because we use a steady-state model, we scale the inferred accumulation rate for the past 4.7 ka to the long-term (past four glacial cycles, 450 ka) average at Dome C using the AICC2012 chronology (Bazin et al., 2013; Veres et al., 2013). The average accumulation rate for the past 4.7 ka is 2.9 cm/yr and for the past 450 ka is 2.1 cm/yr. This gives a factor of 0.72.

Line 114 and below – Model description: It is helpful to present the numerical values of the model parameters (e.g., constants used in the equations) in a table. This would improve clarity and allow readers to reference them more easily.

We have revised the text to point readers to Cuffey and Paterson, 2010 for the temperature dependent values we have chosen. The thermal parameters are given in setup_thermal_herc.m, although we admit they are not that easy to find.

The specific heat capacities values (J/kg/K) are temperature dependent given by Cuffey and Paterson, 2010 as c = 152.5 + 7.122*T.

The thermal conductivity k (W/m/K) is also temperature dependent and we use the values of Waite et al. (2006): k = 8.895 * exp(-0.005182*T).

We have revised the text to give

where T is temperature, z is height above the bed. The thermal diffusivity is given by, $\kappa=K/\rho c$ is thermal diffusivity, where $\rho$ is the constant firn column density based on South Pole values (Lilien et al., 2018), c is the temperature-dependent specific heat capacity given by Cuffey and Paterson (2010; eq 9.1), and k is the thermal conductivity exponential fit to the Waite er al. (2006) data as described in the supplementary information of Buizert et al. (2021) (Specific values can be found in the setup_thermal_herc.m file in the GitHub repository).

Line 121: The value "2 cm/yr" is this in water equivalent or ice equivalent?

This values, and all accumulation rate values, are made in ice equivalent. We have made the following change to indicate that:

The accumulation rates across the study area are inferred from an englacial layer dated to 4.7 ka (Singh et al., in prep) and given in ice equivalent.

Line 147: The Bereiter et al. (2014) model originally comes from Ikeda-Fukazawa et al. (2005), from which all the key parameters are derived. Please cite this work here.

We have added the recommended citation.

Line 150: This part also needs a citation to Ikeda-Fukazawa et al. (2005).

We have added the recommended citation.

Line 154: The correct references are Bazin et al. (2013) and Veres et al. (2013), not 2014.

We have corrected these citations.

Line 154 – 156: In this section as well, the authors should cite the original paper that provides the permeabilities, rather than Bereiter et al. (2014). The manuscript states that only O2 was included in the simulation. However, it is unclear how δO2/N2 values were

calculated—was N2 assumed to remain constant at its initial value? The difference in diffusivity between O2 and N2 is less than one order of magnitude, about a factor of three based on the values from Salamatin et al. (2001). I am not convinced that it is justified to neglect N2 in the simulations. The authors should provide a clear justification or, preferably, a quantitative assessment of the impact of including N2 on the δO2/N2 signal damping. The manuscript states that "including N2 would slightly enhance the signal damping presented in later sections," but I could not find any subsequent section in which the effect of including N2 was actually evaluated or quantified.

One limitation of our diffusion model is the treatment of $N_2$. The model assumes that the $N_2$ content in the ice is uniform, attributing variability in $\delta O_2/N_2$ solely to changes in the $O_2$ concentration. Incorporating variable $N_2$ concentrations and accounting for $N_2$ diffusion would reduce the smoothing effect of $O_2$ diffusion on the $\delta O_2/N_2$ signal, provided that the gradient in $N_2$ concentration has the same sign as that of $O_2$. This condition is likely met as millennial-scale variability in both $\delta O_2/N_2$ and total air content in the ice is thought to be controlled by similar bubble-closure processes. Empirical data support this linkage, with $\delta O_2/N_2$ and total air content showing covariance (e.g. Fujita et al., 2009; Lipenkov et al., 2011). As such, this work likely places an upper limit on the diffusive smoothing of $\delta O_2/N_2$ in ice cores. We note that if the $N_2$ and $O_2$ concentration gradients were of opposite sign, the effects of $N_2$ and $O_2$ diffusion would be additive, and smoothing would be enhanced.

We have edited the text to reflect this information, as well as added a citation to the original paper:

We only simulate the diffusion of $O_2$ because $N_2$ has a permeation coefficient roughly one order of magnitude lower than that of $O_2$ (Salamatin et al., 2001). Modeling $O_2$ diffusion is sufficient for modeling the damping of the $O_2/N_2$ ratio. Including $N_2$ diffusion would reduce the smoothing of $\delta O_2/N_2$ from $O_2$ diffusion, provided that the gas concentration gradient of $N_2$ has the same sign as $O_2$. This condition is likely met as millennial-scale variable of $\delta O_2/N_2$ and total air content in ice is thought to be driven by similar bubble-closure processes. This is supported by empirical data showing $\delta O_2/N_2$ and total air content covariance (Fujita et al., 2009; Lipenkov et al., 2011). Thus, this study provides an upper limit on $\delta O_2/N_2$ smoothing in ice cores. Note that if $O_2$ and $N_2$ gas concentration gradients were of opposite signs, the effects of their diffusion would be additive and thus enhance smoothing.

Start from line 187: This paragraph needs further clarification. Was the ice and heat flow model run over 1.5 Myr to derive average annual layer thickness and temperature every 50

kyr? I understand that the CO2 signal was prescribed with 5/4 cycles and the O2 signal with 5/2 cycles in the first 50 kyr interval, and that the diffusion was simulated under conditions where the annual layer thickness gradually decreases and the temperature increases with depth. Why was a 50 kyr interval chosen?

The ice and heat flow model calculates the temperature-depth and age-depth relations, which are later partitioned into 50 kyr intervals in the gas diffusion model. Nothing is inherently special about 50 kyr intervals, it is simply what we chose to best match Bereiter et al (2014)'s results. Below you will find three figures like Figure 3 in the manuscript. The first is the results when the ice column is partitioned into 25 kyr intervals, the second with 50 kyr intervals, the third with 100 kyr intervals:

[Figure]

There is a slight difference in the resulting diffusion based on the interval length, but it is only marginal.

Line 189: The sentence mentions "Figure 3c and d," but it's unclear why they are cited here.

The intention here was to bring attention to Figure 3c and d as an example of simulating diffusion. We have moved and slightly edited the reference as follows:

Gas concentrations are initialized, and the model simulates diffusion for 50 kyr. Next, gas concentrations are saved, and the model physical parameters are updated using the layer thickness and temperature from the next interval. Then, the simulation continues for another 50 kyr. The process iterates until the last interval is reached (e.g. Figure 3c and d).

Line219: I do not understand why Bazin et al. (2013) and Veres et al. (2013) are cited here. Does this mean that the values in Table 1 were taken from these references?

We use the same EDC parameters used by Bereiter et al. 2014 to best compare with their results so the Bazin and Veres references are not needed. We have reworded the sentence to:

In the first run, we simulate the conditions for EPICA Dome C (EDC) and use parameter values from Bereiter et al. (2014) ("EDC" in Table 1).

Line 221: Similarly, I do not understand why Uemura et al. (2018) and Buizert et al. (2021) are cited here.

We use the ice age data from Uemura et al. (2018) and the borehole temperature data from Buizert et al. (2021) to tune parameter values (accumulation rate, surface temperature, ice thickness, GHF) to fit these datasets. We have moved the citations to better reflect what they refer to:

In the second run, we simulate the conditions for Dome Fuji with parameters values tuned to fit the age (Uemera et al., 2018) and borehole temperature data (Buizert et al., 2021) ("DF" in Table 1).

Table 1: While parameters for EDC are described in Bereiter et al. (2014), please explain more clearly in the text how the values for Dome Fuji (accumulation rate, ice thickness, surface temperature, GHF, p) were determined, and cite appropriate sources.

As described in the above comment, we use the datasets from Uemura et al. (2018) and Buizert et al. (2021) to determine the appropriate parameter values.

Figure 4: The lines with SDR values below 0.1 (corresponding to 0 to ~0.8 Ma) are nearly indistinguishable. Please consider adding a zoomed-in inset for this range to improve readability.

We have added a zoomed-in inset as recommended:

[Figure]

Line 279: The effect of ice thickness is difficult to interpret. While the O2 permeability from Salamatin is more sensitive to temperature than the CO2 permeability, the absence of temperature profiles in the ice sheet makes it difficult to fully understand the results. Please prove a more detailed explanation.

We have edited this section to provide more explanation:

Unlike $CO_2$, ice thickness has a relatively weak effect on the $O_2/N_2$ ratio SDR. In thicker ice columns, we find the impact on temperature slightly dominates the impact on gas concentration gradients as demonstrated by the slightly positive slope in Fig. 6b. This arises from a higher sensitivity to temperature in the permeability of $O_2$ compared to $CO_2$. However, in shallower ice columns, where temperatures are cooler compared to thicker ice columns, the relatively flat slope implies the impacts on temperature and layer thinning are

about equal and opposite. The lower dissociation pressure of $O_2$ compared to $CO_2$ causes more diffusion in the warmer temperatures of thick ice columns.

Figure 8: Please consider improving the color scale. In panel (a), the contrast in accumulation rate is barely visible. In panel (b), why is the color scale reversed compared to Figures 2 and 9?

We have revised the color scales for panels (a) and (b):

[Figure]

Line 341: Please indicate in the figure that where Dome A is.

We feel adding an indicator for Dome A in Figure 8 will make it too busy. We have instead reworded line 341 to reflect the region we define as the "Foothills" in Section 2.1, and the added map to Figure 2 includes Dome A.

Line 396: "Our results show that CO2 SDR for 1.5 Ma ice does not exceed 13 % in the grid-north Foothills and averages 5 % (Figure 8k)." This sentence should include something like "with 50 mW/m² GHF" to clarify the condition of the result shown in Figure 8k.

We have added the recommended phrase as follows:

Our results show that $CO_2$ SDR for 1.5 Ma ice does not exceed 13 % in the grid-north Foothills and averages 5 % given 50 mW m$^{-2}$ GHF (Figure 8k).

Line 400: Did you also test a 20 kyr periodic CO2 signal? If so, please consider including a figure showing the results, either in Figure 3 or elsewhere.

We did not extensively test 20 kyr periodic CO2 signals as the focus of the paper is on 40 kyr CO2 signals due to their relevance to pre-MPT glacial cycles.

We test a small set of 20 kyr periodic CO2 signals in response to a comment from the other reviewer (see that response for more details). With those results at 1.5 Ma, we find 20 kyr periodic CO2 signals have SDRs ~3.5 times higher than CO2 signals with 40 kyr periods. Similarly, O2/N2 signals of 20 kyr periods have SDRs ~1.8 times higher than O2/N2 signals of 40 kyr periods, but with more variability. We also note that we have made our code publicly available (and has been used by the other referee) so that anyone interested can test the period they are interested in.

Section 4.3: The experiment is unclear. In Case 2, does the temperature profile correspond to the upper part (0–2430 m) of the 2700 m case? Similarly, does Case 3 use the profile from 3000 m ice thickness truncated at 2700 m? If so, Case 2 includes both the thinning of the ice and lower basal temperatures, while Case 3 isolates the temperature effect. Please confirm and clarify.

We have added a figure (as detailed in a previous comment) showing the temperature profiles for each case. Note the y-axis now shows depth below the surface, not height above the bed like in Figure 3. We recognize that these cases are difficult to describe in words and have made changes to the text as described below (in response to specific points raised).

Case 2 is to simulate the layer thinning effect a 270 m non-deforming basal ice layer would have on the 2700 m case (Case 1). We chose GHF for Case such that at 2430 m below the

surface, both Case 1 and 2 have the same temperature; this happens to happen with the same GHF in both cases.

Case 3 is to simulate the change in temperature effect a non-deforming basal ice layer would have. We chose GHF for Case 3 such that if Case 1 is extended to 3000 m (i.e., the other parameters are unchanged but total ice thickness increases from 2700 m to 3000 m; we call this Case 1*) then at 2700 m below the surface both Case 3 and Case 1* have the same temperature.

[Figure]

Figure 10: The temperature (left) and age (right) profiles for each model case. The age profiles of Cases 1 and 3 are the same. Note that depth is measured from the surface, not as height above the bed.

See below for text changes.

Line 437: The accumulation rate is stated as 2 cm/yr here, but Table 3 lists it as 3 cm/yr.

The correct value is 2 cm/yr. We have corrected the table caption.

Line 439: From "In Case 2, a thinner deforming…" I suggest inserting a line break to improve readability.

We have added the recommended line break.

Line 440: Case 2 involves not just thinning of the ice but also changes in temperature? Please explain this more clearly.

As described above, we adjust the parameters of Case 2 so that the temperature profiles of Case 1 and 2 align at 2430 m below the surface. Thus, there is no change in temperature. This is so we can isolate the effects of layer thinning in Case 2. In Case 3, we look at how changes in temperature affect gas diffusion, without the effects of layer thinning.

We have edited the paragraph to be more clear in its explanations:

To assess the impact of a potential basal ice layer, we perform three model runs and compare their SDRs. Case 1 is the control with 2700 m ice thickness, 50 mW m$^{-2}$ GHF, 2 cm yr$^{-1}$ accumulation rate, and -60 ℃ surface temperature. Case 2 investigates the impact of a change in layer thickness while keeping the temperature of the ice sheet the same. Case 3 investigates the impact of a change in temperature while keeping the layer thickness the same.

The temperature and age profiles of the three cases are shown in Figure 10. We do not directly incorporate a non-deforming basal layer and instead adjust the GHF to simulate its impacts. Case 2 simulates a 10 % (270 m) basal ice layer to assess the impact of layer thinning from a non-deforming basal ice layer; we model a thinner ice column (2430 m) but match the Case 1 ice temperature. Case 3 simulates a 3000 m ice column with the bottom 10 % (300 m) non-deforming to assess the impact of a change in ice temperature from a basal ice layer; we model the same thickness (2700 m, so the depth-age profile is the same), but match the temperature to a model run with 3000 m ice thickness by reducing the GHF (Case 1* in Fig. 10). The cases and resulting SDR are summarized in Table 3.

Line 452: Oyabu et al. (2021) demonstrated that the permeability coefficients proposed by Salamatin et al. (2000) reproduced the smoothing of the O2/N2 signal in the Dome Fuji core well. However, their simulations were conducted over a limited temperature range. The temperature dependence of Salamatin's permeability is quite strong, with estimates indicating that the permeability increases by approximately one order of magnitude for every 10°C increase. In contrast, the "Fast set" proposed by Ikeda-Fukazawa et al. (2005) exhibits behavior that approaches the "Slow set" at higher temperatures, such as near the base of the ice sheet.

While the reliability of these permeability estimates remains uncertain, if one takes an optimistic view, it could be argued that the use of Salamatin's coefficients results in a more conservative estimate of signal preservation near the base of the ice sheet. Future measurements on actual ice cores may help clarify the temperature dependence of these permeability coefficients. It may be worth mentioning these points in the discussion.

We have added the following to the text to reflect these ideas:

Oyabu et al. (2021) have previously demonstrated that $O_2$ permeation values an order of magnitude faster than those we consider here yield unrealistically high smoothing compared to $O_2/N_2$ measurements from the Dome Fuji core. Those researchers also show that the permeation coefficients of Salamatin et al. (2001) yield diffusive smoothing in reasonable agreement with the Dome Fuji measurements, increasing confidence in our parameterizations. However, the permeation values provided by Salamatin et al. (2001) (the "slow set") are tested over a limited temperature range, and the values given by Ikeda-Fukazawa et al. (2004) (the "fast set") approach the values in the "slow set" at higher temperatures, like those near the bed of the ice sheet. Thus, our use of the "slow set" in this study may provide a more conservative estimation of diffusive smoothing of $O_2/N_2$. Future ice core measurements may improve our understanding of the temperature dependence of these permeation coefficients.

Line 518: Adam Auton (2024). Red Blue Colormap (https://www.mathworks.com/matlabcentral/fileexchange/25536-red-bluecolormap), MATLAB Central File Exchange. Retrieved November 18, 2023.

This is not cited in the text.

We have added a citation for this (and other utilized MATLAB packages) in the Code Availability section.

Figure 1: The manuscript mentions "AICC2023," but it is unclear whether this refers to the use of the AICC2023 age scale for the ice core chronology.

This refers to the O2/N2 data from this paper. We have revised the citation to Bouchet et al. (2023) instead.

---

## Author Comment (AC2)

Thank you to Thomas Bauska for providing specific and valuable feedback on our manuscript. We have carefully reviewed and incorporated the recommendations into a revised manuscript and describe the changes in the following response. The reviewer's responses are written in black text, and our answers are written in red text. Revised sentences in the manuscript are written in blue text.
* * *
Review of Sailer et al., Ice Core Site considerations from modelling CO2 and O2/N2 ratio diffusion in interior East Antarctica.

*Sailer et al.,* have performed a critical analysis of the potential preservation of gas signals at locations in Antarctica containing the world's oldest ice with implications for existing and future ice coring efforts. I really enjoyed reading this study and particularly appreciated that the authors provided their code (which I have tested and look forward to using in the future!). By and large, the authors employ a previously established methods (Bereiter et al., 2014) and existing data on gas diffusion (Ahn et al, 2008; Ikeda-Fukazawa et al., 2005, etc) but expand the scope of previous work to include new, unexplored regions of Antarctica. In this effort, they also expand the range of parameters tested by Berieter et al., 2014 to investigate the sensitivity of their model to different temperatures, accumulation rates, geothermal heat flux, ice column thickness, etc. – effectively all the key parameters one would want to see varied when looking for old ice sites.  I would support publication in the Climate of the Past after some revisions, mostly minor.

We thank Dr. Bauska for his enthusiasm and helpful suggestions. We are particularly glad he went to the trouble to run our code and test its performance.

First, I will raise a few bigger picture questions and then go line-by-line.

**Can this analysis be reconciled with the observations that there is O2/N2 variability in 1.5-million-year-old ice at Allan Hills?** The major take-home from both Bereiter et al., 2014 and this study is that we should have lost all the 20-kyr and shorter variability.  Yet Yan et al., 2021 report variations in O2/N2 from very old ice at Allan Hills. I'll admit that you'll be comparing apples-to-oranges when jumping from a well-resolved deep ice core to a jumbled-up blue ice site where the ice (at least in the present) is colder than the ice buried

~3,000 meters below East Antarctica.  None-the-less, I can't reconcile in my mind how O2/N2 variations would remain if the diffusion rates are indeed so fast. In fact, I even tried a quick test with your model by making an isothermal ice column at -30C.  The SDR of O2/N2 @1.5 million years age (with about 10,000 years per meter) was coming out quite high at 0.86.

This is an interesting observation and one that we have pondered too. There are many unknowns of the Allan Hills ice, which make it difficult to assess how much diffusion should occur in Allan Hills ice.

First, we do not understand the ice flow history of the old ice parcels. Where and how thick was this ice in the past? When did it thin to its current thickness? We have few constraints on these questions.

Second, the Allan Hills ice cores are not continuous climate records. Mechanical mixing (as indicated by the age-scale reversals) could tend to homogenize the O2/N2 ratios, but might also bring packets of ice that have been isolated from each other into contact more recently than the age of the ice. We note that emerging work, particularly improved dating, from COLDEX is suggesting that the dO2/N2 trend in older ice could be related to age, rather than insolation.

Third, we don't know the starting conditions very well. The S27 core provides some reasonable estimates, but the firn densification processes in a 40ka world may be different. Further, the Allan Hills firn is challenging to understand because the accumulation rates are very low, and potentially switch from net accumulation to net ablation.

Because of the large uncertainties in the histories and processes, we do not want to include such speculation in this manuscript. This is a good topic for future work. One thought that occurs to us in response to this comment, is that we might be able to use high resolution dO2/N2 measurements from the Allan Hills to better constrain the diffusivity, which is something we will discuss within the project at an upcoming meeting.

On a somewhat related note, from Sailer et al. it's not as clear as in Berieter et al., 2014 if we will have lost longer periods variation. Once limitation of this study is they only consider CO2 changes on the obliquity timescale and O2/N2 on the precessional timescale.  I understand why the authors have chosen to impose the two different timescales to illustrate potential gas preservation.  On the other hand, it makes for a somewhat convoluted apples-to-oranges comparison for the reader during some later stages in the paper.  I hesitate to call for major revision, so I will only loosely suggest to the authors that

they consider running experiments with the same timescale of forcings for both gases. One could then derive a parameter that describe the additional smoothing of O2/N2 compared to CO2. From the gas world perspective, yes, we want to know the absolute smoothing. But we'd also want to know if the ratio of smoothing between gases changes with under different conditions.

We have investigated the amount of diffusion for each gas given the same period of variation. We overall expect O2/N2 to diffuse more than CO2, although at the warmest temperatures the permeability of O2 increases above that of CO2, which may affect the relative diffusion. Because the permeabilities have different temperature sensitivities, we expect complexity in the relative diffusion of the two gases. To see to what extent the O2/N2 SDR is fixed relative to the CO2 SDR, we test a parameter space with variations in each of the four forcings: 2, 2.5, 3 cm/yr accumulation rate; -60, -57.5, -55 C surface temperature; 2500, 2750, 3000 m ice thickness; 45, 47.5, 50 mW/m$^2$ GHF. We then calculate the ratio of O2/N2 1.5 Ma SDR to CO2 1.5 Ma SDR. If the ratio is constant, then O2/N2 always diffuses by the same amount more than CO2.

What we find, shown in the figure below, is that on average O2/N2 diffuses 3.6 times more than CO2 on 20 kyr period variations, and 7 times more on 40 kyr period variations. There is noticeable variation too. The ratio of O2/N2 SDR to CO2 SDR generally decreases as CO2 SDR increases, but this is largely because O2/N2 SDR reaches an upper limit of 1, although the higher O2 permeability at warm temperatures may also be playing a role. For both period variations, the parameter combinations that lead to the smallest ratio between SDRs (i.e. where O2/N2 has diffused relatively less compared to CO2) are where accumulation rate is high and ice thickness is low, which leads to more layer thinning. The opposite is also true; the SDR ratios are greatest where accumulation rate is low and ice thickness is high, which leads to less layer thinning. This makes sense since we find in Section 3.2 that O2/N2 is more sensitive to the temperature changes in the thicker ice scenarios than CO2.

Note that while the figure is ordered by increasing CO2 SDR, the O2/N2 SDR for each parameter combination does not have the same order. The high frequency variations in the ratio are an indication of this. For instance, the CO2 SDR can remain unchanged, while the O2/N2 SDR increases. The conditions for such an occurrence can be complicated, with the CO2 SDR is remaining unchanged due to offsetting impacts, such as thicker ice creating both less layer thinning and also warmer temperatures. These impacts can then affect O2/N2 differently, in this case the warmer temperatures having a larger impact than the thicker layers. In the manuscript, we have explored related ideas with the sensitivity

analysis (Section 3.2). Although this does not directly compare the two gases with the same period of variation, the sensitivity to various forcings is independent of the period.

After this analysis, we chose not to include this in the paper because we did not find a clear message and believe that adding a discussion of this in the manuscript would distract from the main focus. The most consistent trend we find in this experiment is driven by O2/N2 fully diffusing; otherwise, the relationship of diffusion of O2/N2 and CO2 is complex. We hope that the code provided will allow readers interested in technical questions, such as this one, the ability to explore them.

[Figure]

**How does this study differ from Berieter et al., 2014 (if it all) in approach?** From my reading, I believe the temperature-depth-age models are virtually indistinguishable. However, this would be nice to be spelled out exactly. Particularly if there are a few differences.

The age model is identical to the one in Bereiter et al. (2014). The temperature model differs in that we use temperature dependent thermal conductivity and specific heat capacity instead of ice column averages. We have edited the text to reflect this:

We use a one-dimensional, steady state ice and heat flow model to calculate the temperature and age of ice with respect to depth. The age model is identical to that in Bereiter et al. (2014). The temperature model differs in that we utilize a temperature dependent thermal conductivity and specific heat capacity, rather than an ice column average.

**A few more tests to confirm performance of the model would be useful.** I recommend showing at least two examples of how the model performs for both a low-accumulation site with little melting (Dome Fuji?) and a high-accumulation site with high melting (WAIS Divide)? In both cases, I believe the borehole temperature and melt-rates (possibly inferred not measured directly) data should be available. It would be sufficient to this only for the review documents or in a supplemental figure. I don't think it would be necessary for the main body. That said, the introductory material is somewhat lacking in a real-world grounding that could be better illustrated for non-ice core specialist. For example, around lines 55-60, it's taken for granted that reader has a good grasp how thinning and temperature evolve with depths. This could benefit from an illustrative figure that shows some real-world examples. For example, the way Berieter et al., 2014 introduces the problem with real and modelled examples in Figure 1 is quite useful.

To better assist the reader in understanding temperature and thinning, we have included additional panels in Figure 3 to show what temperatures ice packets of different ages are experiencing and how thin the layers have become. Additionally, modeling locations with high accumulation rates, such as WAIS, is not feasible with the steady-state model. We discuss this further in a later response.

Also, on the subject of model testing, I took my best crack at an apples-to-apples comparison with a similar model we use in-house at BAS and found very good agreement in the modelled temperature and age with depth. Suffice it to say, the version of the model I used is mostly educational purposes such as teaching "where to find old ice?" exercises so a full model inter-comparison isn't needed and well beyond the scope of this review. However, I noticed one discrepancy could call for a little bit more description of your model. In the scenario with -60C surface temperature, *2 cm a-1 a* accumulation and 3,000-meter thick ice column, I didn't see any basal melting until the geothermal heat flux tipped over 55 mW a-1.

We get melt initiating 55 mW m$^{-2}$ with these forcings, assuming p=4 for the vertical velocity profile. We aren't sure how to further investigate this. In terms of differences between the BAS model and ours, there could be a variety of differences such as temperature-dependent thermal properties and a constant firn column based on South Pole values.

*Important aside, is that water or ice equivalent accumulation?

Accumulation is in ice equivalent. This has been addressed in the following change:

The accumulation rates across the study area are inferred from an englacial layer dated to 4.7 ka (Singh et al., in prep) and given in ice equivalent.

Upon further investigating and the running the code for your model I saw that the temperature of the bedrock (possibly down to a few kilometers is modelled). This is more sophisticated than the model I used and probably the main reason for the difference. **It would be nice to know a little bit more about this portion of the model, particularly as the areas you eventually rule out for old ice exploration appear sensitive to the presence or absence of melting.**

This steady-state ice-and-heat flow model is actually what was used to initialize the transient model which has been used to infer the magnitude of LGM-Holocene temperature change at Dome C and Dome Fuji (Buizert et al., 2021) and in other work like tracking ice parcel temperatures at WAIS Divide (Aydin et al., 2014) or calculating geothermal flux constraints at ice rises (Fudge et al., 2019). Modeling the bedrock is important for transient applications but isn't critical for steady-state applications. We have added more description to the text, as detailed in our next response.

Also, I struggled a bit to understand the iterative approach to solve the basal melt rate. I believe a bit more detail is warranted along with a brief review of the implications of this approach. For examples, could you calculate how much heat is "lost" via conduction up into the ice and how much is "lost" via latent heat? Also, I assume the implication is that the latent heat is indeed "lost" from the system? As in, it is implied that a thin layer of water is flowing away from the site?

The basal melt needs to be calculated iteratively because applying a basal melt rate affects the vertical velocity (equation 2) and hence the englacial temperature profile. We revise our description of the basal melt to:

The deformation parameter is assumed to be p = 4 in all scenarios, unless otherwise stated. We include bedrock in the thermal calculation and set the GHF at 7 km below the ice surface. The melt rate, m, is first set at 0 for calculating the temperature profile. Equation (1) is solved similarly to Firestone et al. (1990) with an integration factor ...

The melt rate is calculated from the excess geothermal heat that the ice cannot conduct away from the bed. This is calculated iteratively because the melt rate affects the vertical velocity (equation 2) and thus the englacial temperature profile. The GHF is equal to the

heat conduction through the basal ice and the latent heat lost in melting, which we assume is lost as the water flows away at the bed.

Finally, to circle back to my original suggestion of doing some model-data comparison, the "proof would be in the pudding". So if the model does well at simulating the temperature and melt-rates at Dome Fuji and WAIS Divide (or any other sites of your choosing) then the reader will be more confident in the approach.

Because the model is steady-state, a model-data comparison is a bit harder than it might seem. For a site like WAIS Divide, whose mid-depth temperatures are colder than the surface temperature, the ice sheet temperatures clearly retain the thermal signature of the LGM (Cuffey et al., 2016). A steady-state model cannot capture the temperature profile. For sites like Dome C and Dome Fuji, a steady-state model does a better job, but even then we've shown that the borehole temperature retains information that can be used to estimate the LGM cooling (Buizert et al., 2021). The transient temperature influence is largest at the surface and minimal near the bed, which is why our assumption of steady-state temperature is appropriate for this study – the temperature of the old ice does not vary much. But the change in temperature near the surface makes a model-data comparison difficult. We illustrate this with the following figure for Dome Fuji where we run the steady-state model under four forcing scenarios. The first three have either a modern, LGM, or glacial-interglacial-average surface forcings and a geothermal flux of 50mWm-2 which puts the base at the pressure melting point without inducing much melt. The fourth scenario uses the glacial-interglacial average surface forcings, but a lower geothermal flux.

[Figure]

[Figure]

We want to highlight two take-aways from this figure

1 The borehole temperature profile is fit decently (i.e. within a couple of degrees) by the modern surface forcing but the mid-depths are too warm, which is because the colder glacial temperatures are not part of the history. The LGM and glacial-interglacial average surface forcings yield too cold of temperatures in the upper third of the ice sheet, but improve the fit in the lower two thirds. The temperature variations in upper ice sheet are not particularly important for our analysis for two reasons: a) the layers are thick so little diffusion occurs, and b) the temperatures are cold so little of the overall diffusion occurs.

2. The temperature misfit is much larger if the basal temperature is incorrect. The transient temperature variations are small compared to not getting the basal forcing correct. This is additionally important because most of the diffusion occurs in the warm, highly strained, basal ice which does not vary temperature much due to the time-varying surface temperature.

**All the maps could be made more accessible.** I suggest you provide some more geographic context as I was a little bit lost as to the extent of the "COLDEX Search Region". I suggest adding an Antarctic-wide map inset to Figure with the region currently covered in Figure 1 highlighted. Dome A and Dome C play important roles in the paper. Please show their location and also an arrow pointing in their direction (also Vostok?). The subplots in Figure 8 (as long as they are same extent as Figure 1) are okay but seem to be missing lines of latitude and longitude. I would also appreciate some more points of reference here.

We have added a map to Figure 2 to better present the geographic context:

[Figure]

**Line-by-line comments:**

Abstract: "foothills" is not yet precisely defined and comes across as quite a colloquial term to use in the abstract. I recommend using a more precise term, or using the spatial information you provide about location between Dome A and South Poler, or add on a line like "...roughly equidistant between Dome A and South Pole, a region we call the "Foot Hills" of the Gamburstev Mountains"

We have revised the sentence as follows:

The most promising region for recovering 1.5 Ma ice is approximately 400 km from both South Pole and Dome A, a region we call the "Foothills," due to low accumulation rates and moderate ice thickness.

Line 35: *"However, this method is limited, providing different results based on the species and location and requiring several assumptions. Köhler (2023) suggests that some of these assumptions may be incorrect by comparing reconstructions to a carbon cycle model."* It would be helpful for a reader unfamiliar with Kohler et al to mention some of these key assumptions. Otherwise, it sounds like a quite a broad, unsupported swipe at the boron method.

We have added some of the assumptions Köhler (2023) discusses:

However, this method is limited, providing different results based on the species and location and requiring several assumptions. Köhler (2023) suggests that some of these assumptions, such as equilibrium between atmospheric and equatorial sea surface $pCO_2$, lower estimated surface ocean pH in the Pacific than the Atlantic, and assumptions on total alkalinity and dissolved inorganic carbon, may be incorrect by comparing reconstructions to a carbon cycle model.

Lines 55-65. There are some very short, two-sentence, paragraphs here. I think it could be restructured into one, possible with some bullet points.

We have condensed these points into one paragraph by removing some line breaks.

Line 74: *"This is the most reliable method for dating ice cores of this age."* A fairer statement would be that O2/N2 is one of the key pillars of dating as in practice all available information is used (e.g. Bouchet et al., 2023)

Agree, we have reworded the sentence as follows:

This is a key method in dating gases in the oldest ice cores.

Line 150. *"Bubbly ice resides in the upper ~1000 m where diffusive smoothing is unimportant due to thicker annual layers and colder temperatures."* I would rephrase. As shown by experimental evidence that underlies these results (Ahn et al., 2008) diffusion dues occur in bubble ice. In fact, one could argue that those rates derived in Ahn and most subsequent work don't apply for clathrate ice in question here. I would say something like "although diffusion occurs within bubbly ice, the time spent within the bubble phase is relatively short (e.g. 25,000 years at a site like EDC) compared to the timescales of interest here (e.g 1,500,000)."

We have made the following changes to reflect this suggestion:

While diffusion does occur within bubbly ice, the time gases spend in the bubble phase is relatively short compared to the timescales of interest here (e.g. ice 1000 m below the surface at EDC is ~65 ka, roughly the extent of bubbly ice; Bazin et al., 2013; Veres et al., 2013).

Please provide some more introduction to "fast" and "slow" datasets. Why the large discrepancy? I wouldn't expect the authors to solve the problem but a figure like presented in Bereiter et al, 2014 would be helpful. I found myself going back and forth between Bereiter et al., and this study quite a lot.

We have added additional context for the fast and slow sets and a supplemental figure based on Figure 2 in Bereiter et al. (2014):

Bereiter et al. (2014) modeled $O_2/N_2$ diffusion under two sets of gas parameters, a "fast set" and "slow set." The "slow set" parameters are derived from empirical fits to ice core data (Salamatin et al, 2001). The "fast set" parameters are based purely on molecular dynamics simulations of gas diffusion in ice (Ikeda-Fukuzawa et al., 2004).

[Figure]

Table 1: I know it is mentioned somewhere the text, but it should be reiterated that you appear to be using some sort of glacial-interglacial average for the sites. Otherwise, -60C at EDC jumps out as the reader as strangely cold.

We have changed the caption to the following:

Forcing values used for each model scenario. EDC surface temperatures approximate a glacial-cycle average. The SDR from 0 to 1.5 Ma of each scenario is shown in Fig. 4.

Figure 4. *"Current ice core measurements cannot be used to estimate diffusion in older ice."* I would disagree with such an unequivocal statement. One take home (I had) from Bereiter et al., 2014 is that millennial-scale and faster variability can be used to estimate diffusion. A narrower statement, like "current orbital-scale variations in gases cannot be used to estimate diffusion" would be more apt.

This text has been removed from the figure caption and additional explanation has been added to the main text:

It is important to consider gas diffusion in older ice because the preservation of orbital-scale variations in current ice cores does not imply that orbital-scale variations will persist in ice nearly twice as old.

Line 405: Does TAC diffuse? This is an interesting point.  Actually, I believe it could be easily tested with your model as *Uchida et al., 2011* provide an estimate of whole air diffusion. They are shown in a figure in Bereiter et al., 2014.

TAC diffusion is an interesting topic for the future, but ultimately beyond the scope of this paper. Additionally, we are unsure about the values presented in Uchida et al. (2011), particularly at low temperatures.

Overall, very nice work.  I'm looking forward to seeing the revision and then the paper published.  Also, I'm excited to use this model!

All the best,

Thomas Bauska

Uchida T, Miyamoto A, Shin'yama A, Hondoh T. Crystal growth of air hydrates over 720 ka in Dome Fuji (Antarctica) ice cores: microscopic observations of morphological changes below 2000 m depth. *Journal of Glaciology*. 2011;57(206):1017-1026. doi:10.3189/002214311798843296

Yuzhen Yan *et al*. Ice core evidence for atmospheric oxygen decline since the Mid-Pleistocene transition.*Sci. Adv.*7,eabj9341(2021).DOI:10.1126/sciadv.abj9341